# Underlying Mechanisms Involved in Gambling Disorder Severity: A Pathway Analysis Considering Genetic, Psychosocial, and Clinical Variables

**DOI:** 10.3390/nu15020418

**Published:** 2023-01-13

**Authors:** Neus Solé-Morata, Isabel Baenas, Mikel Etxandi, Roser Granero, Manel Gené, Carme Barrot, Mónica Gómez-Peña, Laura Moragas, Nicolas Ramoz, Philip Gorwood, Fernando Fernández-Aranda, Susana Jiménez-Murcia

**Affiliations:** 1Behavioral Addictions Unit, Department of Psychiatry, Bellvitge University Hospital, Feixa Llarga s/n, 08907 L’Hospitalet de Llobregat, 08907 Barcelona, Spain; 2CIBER Physiopathology of Obesity and Nutrition (CIBERObn), Instituto de Salud Carlos III, 28029 Madrid, Spain; 3Psychoneurobiology of Eating and Addictive Behaviors Group, Neurosciences Programme, Bellvitge Biomedical Research Institute (IDIBELL), 08908 Barcelona, Spain; 4Department of Psychiatry, Hospital Universitari Germans Trias i Pujol, IGTP Campus Can Ruti, 08916 Badalona, Spain; 5Department of Psychobiology and Methodology, Autonomous University of Barcelona, 08193 Barcelona, Spain; 6Genetics Laboratory, Legal Medicine and Toxicology Unit, Public Health Department, Faculty of Medicine, University of Barcelona, 08907 Barcelona, Spain; 7Institute of Psychiatry and Neuroscience of Paris (IPNP), INSERM U1266, Team Vulnerability of Psychiatric and Addictive Disorders, Université de Paris, 75014 Paris, France; 8Department of Clinical Sciences, School of Medicine and Health Sciences, University of Barcelona, 08907 Barcelona, Spain

**Keywords:** gambling disorder, severity, neurotrophic genes, socio-demographics, personality traits, psychopathology

## Abstract

Gambling Disorder (GD) has a complex etiology that involves biological and environmental aspects. From a genetic perspective, neurotrophic factors (NTFs) polymorphisms have been associated with the risk of developing GD. The aim of this study was to assess the underlying mechanisms implicated in GD severity by considering the direct and mediational relationship between different variables including genetic, psychological, socio-demographic, and clinical factors. To do so, we used genetic variants that were significantly associated with an increased risk for GD and evaluated its relationship with GD severity through pathway analysis. We found that the interaction between these genetic variants and other different biopsychological features predicted a higher severity of GD. On the one hand, the presence of haplotype block 2, interrelated with haplotype block 3, was linked to a more dysfunctional personality profile and a worse psychopathological state, which, in turn, had a direct link with GD severity. On the other hand, having rs3763614 predicted higher general psychopathology and therefore, higher GD severity. The current study described the presence of complex interactions between biopsychosocial variables previously associated with the etiopathogenesis and severity of GD, while also supporting the involvement of genetic variants from the NTF family.

## 1. Introduction

Gambling is a ubiquitous and generally acceptable activity in our society [1]. Although most people gamble without suffering health issues, some individuals develop gambling disorder (GD) [2]. According to the *Diagnostic and Statistical Manual of Mental Disorders* (DSM-5), GD is an addictive disorder characterized by recurrent gambling that leads to severe psychological, social, and economic consequences [3]. Moreover, recent changes in availability, promotion, and legislation of gambling activity have resulted in an unprecedented growth of the gambling industry, also accompanied by remarkable increases in the GD prevalence [4,5]. In Europe, GD prevalence is up to 3% while in North America and Asia it increases to 5% and 6%, respectively [6,7]; it is therefore defined as a major public health issue which needs to be properly addressed [8].

Even though GD is a relatively recently recognized mental disorder, several risk factors have already been identified that involve individual or biological variables and environmental factors [9]. For instance, cultural background and gambling availability, as well as socio-demographic characteristics (i.e., being single and having low socioeconomic and educational levels) of the individuals who gamble play an important role in the development of the disorder [1,10]. Similarly, being male [11,12] and young [10,13] have been classically considered individual risk factors for GD. Moreover, compared to women, men present an early GD onset [14,15], although the time between the onset of gambling activity and the development of gambling problems appears to be shorter in women (i.e., telescoping effect) [16,17]. Likewise, both males and younger people have preferences for strategic gambling where the gambler’s skills play a role in the result of the gambling activity regardless of chance (e.g., casino, cards, sports betting) [9]. Noticeably, they usually bet higher amounts of money than their counterparts [18], which previous literature has linked to the severity of the disorder [19]. In contrast, women and older individuals are characterized by non-strategic gambling preferences (e.g., bingo, lotteries, slot machines) and higher frequency of comorbid general psychopathology (e.g., anxiety, depressive symptoms) [20]. In fact, a worse psychological state has been associated with the severity of gambling behavior among people with GD [21].

In this line, psychological variables have also been of interest and several studies have found that impulsivity is a nuclear characteristic of addictive-related disorders [22]. Indeed, higher scores on impulsivity measures and related constructs such as personality traits (i.e., sensation seeking) have been described in GD [22] that are, overall, linked to younger age, gambling preferences, and higher GD severity [23,24,25]. Individuals with high impulsivity are usually younger males with preferences for strategic gambling, whereas characteristic traits for individuals with a personality profile defined by high harm avoidance tend to be female sex, older age, higher emotional vulnerability, and non-strategic gambling preferences [20,26]. Taken together, these findings point to the fact that GD represents a phenotypically heterogeneous disorder [27,28].

From a genetic perspective, several studies have shown that inherited factors account for approximately 50% of the risk for GD [29,30]. Hence, genetic mechanisms underlying GD onset, maintenance, and severity are of particular interest. Before genome-wide association studies (GWAS) were made possible in GD, molecular genetic studies applied candidate gene approaches, mainly reporting an involvement of neurochemical systems, such as the dopaminergic and serotonergic systems [9]. Although GWAS have made considerable progress towards an understanding of many complex diseases [31], the single case–control GWAS study in GD did not identify significant regions associated with the disorder. However, an association between a polygenic risk score for alcoholism and severity of problem gambling was reported, which also supported the idea of a link between different psychiatric disorders, such as addictive-related disorders (e.g., substance use disorders and GD), based on shared biopsychosocial vulnerability features [32,33].

Searching for potential genetic targets, previous studies have associated neurotrophic factors (NTF) with the pathophysiology of neuropsychiatric disorders [34,35]. Precisely, a recent study by our group [36] analyzed the involvement of NTF genetic variants in the vulnerability of developing GD. As interesting results, some genetic polymorphisms related to neurotrophin 3 (NTF3) and the BDNF’s tyrosine kinase receptor type 2 (NTRK2) genes were significantly related to a higher risk for GD. 

The dysfunctions found in these genetic variants may have endocrine implications since the expression of the corresponding endogenous ligands would be altered and would therefore imply changes in normal brain signaling cross-talk. Specifically, NTF3 develops neurogenetic and neuroprotective functions in dopaminergic and noradrenergic neurons [37] that are involved in addiction and rewards pathways [38,39]. Meanwhile, NTRK2 binds the brain neurotrophic factor (BDNF), which has been the most studied NTF among addictive-related disorders to date [34,40,41]. In GD, an increase in its circulating concentrations has been described, as well as an association between BDNF concentrations and GD severity [42,43,44,45]. At the same time, NTRK2 seems to be binding not only BDNF but also NTF3 [46]. These findings support the involvement of the NTF family in the pathophysiology of GD at both genetic and endocrine levels. 

Going one step further, clinical implications of these findings should be highlighted since genetic and endocrine dysfunctions in the complex NTF3 and its receptor (i.e., NTRK3), as well as in NTRK2 and their targets (i.e., BDNF and NTF3), have also been reported in affective disorders, attention deficit/hyperactivity disorder (ADHD), and eating disorders (EDs), among others [47,48,49,50], which are not infrequently comorbid conditions with GD [9,51]. With regard to the NTF family, these findings reinforced the idea of common vulnerability factors among different psychiatric disorders, which may be underlying transdiagnostic features, such as impulsivity [52,53].

At this stage, there has been a broad consensus that GD is a complex and heterogeneous disorder, and several phenotypic profiles of vulnerability have been identified, which also influence its severity [28,54]. Furthermore, it has also been proposed that GD probably relates to small genetic contributions in affected individuals’ interaction with other biopsychosocial variables [9]. While the identification of risk factors associated with the disorder has been crucial so far, a more comprehensive approach to the multifactorial interplay (i.e., genotypic, and phenotypic factors) that underlies not only the development but also the severity of GD would be truly valuable for clinical application. 

Characterizing GD severity phenotypes and endophenotypes could also allow clinicians to design more personalized preventive and therapeutic interventions aimed at modifying the course of the disorder since early clinical stages, with a special focus on those people with more vulnerable characteristics. Furthermore, since research based on the biological basis of GD is underexplored and no pharmacological treatment is officially approved in GD to date, a growing body of knowledge related to biological mechanisms involved in GD could help to elucidate new biological therapeutic targets, such as those related to the NTF family. 

Regrettably, the complex relationship between genotypic and phenotypic features (e.g., genetic, psychological, clinical, and socio-demographic factors, etc.) means that the association between genotypes and phenotypes is not straightforward. To the best of our knowledge, this is the first clinical study that examined potential interactions between genotype and phenotype in GD with the aim of delineating profiles of vulnerability, which could also predict GD severity. Regarding genetics, NTF genetic variants significantly associated with an increased risk for GD were used [36]. Other biopsychosocial variables were also assessed, namely socio-demographic features (i.e., sex, chronological age, civil and employment status, educational level), personality profile, general psychopathology, and some characteristics of the gambling behavior (i.e., age of GD onset, GD duration, gambling preferences, gambling activity, debts, bets). For that purpose, whereas classical approaches only allow researchers to test for genotype-phenotype associations, structural equation models (SEM) are a powerful tool to model complex interactions between risk factors and consider the direct and indirect (mediational) links between a broad set of biopsychosocial variables.

Bearing all this in mind, we hypothesized the existence of interactions between genetic polymorphisms and GD severity. Therefore, the presence of certain genetic variants would not only predict the presence of the disorder [36] but also its severity. However, considering that GD is a complex multifactorial entity, we also envisaged a mediational role of other biopsychosocial factors in this interaction between genetics and GD severity. 

## 2. Materials and Methods

### 2.1. Participants and Procedure

The sample was composed of 146 adult outpatients with GD linked to the Behavioral Addictions Unit in the Department of Psychiatry at the Bellvitge University Hospital (Catalonia, Spain). All patients in this study fulfilled the DSM-5 criteria for GD [3]. The recruitment took place between January 2005 and June 2006 [36]. They were evaluated at the Behavioral Addictions Unit in the Department of Psychiatry at the Bellvitge University Hospital (Catalonia, Spain). The assessment consisted of two pre-treatment sessions. In the first session, a semi-structured clinical interview [55] was conducted by experienced psychologists and psychiatrists with a large clinical and research trajectory in the field of behavioral addiction such as GD. In the second session, psychometric assessments and biological samples to analysis genetic variables were obtained. We received completed clinical assessments and biological samples analysis from all the participants included in this study.

### 2.2. Clinical Measurements

*Diagnostic Questionnaire for Pathological Gambling According to DSM criteria* [56]; Spanish validation [15]. This is a self-report questionnaire with 19 items, coded in a binary scale (yes–no), which is used for the GD diagnosis regarding DSM-IV-TR [57] and DSM-5 [3] criteria. In our sample, the internal consistency was adequate (Cronbach’s alpha α = 0.81).

*South Oaks Gambling Screen (SOGS)* [58]; Spanish validation [59]. This questionnaire assesses cognitive, emotional, and behavioral aspects related to problem gambling by measuring the severity of gambling activity (responses ranging from 0 to 20). With 20 items, it allows for the differentiation between non-problem gambling (from 0 to 2), light problem gambling (from 3 to 4), and problem gambling (from 5 to 20, with higher scores being indicative of higher gambling severity). In our study, the internal consistency was adequate (α = 0.79). 

*Symptom Checklist-90 Items Revised (SCL-90-R)* [60]; Spanish validation [61]. This is a self-report questionnaire with 90 items that explores psychological distress and psychopathology using 9 symptomatic dimensions: somatization, obsession-compulsion, interpersonal sensitivity, depression, anxiety, anger-hostility, phobic anxiety, paranoid ideation, and psychoticism. It also includes three global indices: Global Severity Index (GSI), Positive Symptom Total (PST), and Positive Symptom Distress Index (PSDI). In our sample, the internal consistency was excellent (α = 0.98). 

*Temperament and Character Inventory-Revised (TCI-R)* [62]; Spanish validation [63]. Personality traits are evaluated according to seven personality factors that are divided into four factors for temperament (sensation seeking, harm avoidance, reward dependence and persistence) and three for character (self-directedness, cooperation, and self-transcendence). This questionnaire consists of 240 items. The internal consistency in our study ranged between α= 0.71 (novelty seeking) and α= 0.85 (persistence). 

### 2.3. Other Variables

Additional socio-demographic (i.e., sex, chronological age, civil status, educational level, and employment status) and clinical variables related to gambling (i.e., age of GD onset and GD duration, type of gambling which motivated seeking-treatment, and type of gambling modality regarding the preference for strategic gambling, non-strategic or mixed, the presence of debts, and maximum bets) were measured within the first pre-treatment evaluation session [64]. 

### 2.4. Genetic Information

Since this study is a continuation of a previous work by our group, genetic data analyzed in the present study came from the analysis performed by [36]. Briefly, single nucleotide polymorphisms (SNPs) of several NTF genes (nerve growth factor (NGF) gene and its receptor (NGFR), neurotrophic tyrosine kinase receptor type 1 (NTRK1), type 2 (NTRK2) and type 3 (NTRK3), BDNF and neurotrophins 3 and 4/5 (NTF3, NTF4), ciliary neurotrophic factor (CNTF), and its receptor (CNTFR) were selected and genotyped as previously described by Mercader, Saus et al. [50]. Overall, 183 SNPs were genotyped using the SNPlex Genotyping System (Applied Biosystems, Foster City, CA, USA) at the genotyping facilities of CeGen in the Barcelona Node (Centro Nacional de Genotipado, Genoma España). Of the whole available sample, genotyped SNPs which had a call rate lower than 80%, were outside the Hardy–Weinberg equilibrium (HWE), or were monomorphic, were not considered for further analyses (*n* = 25). 

### 2.5. Statistical Analyses

Statistical analysis was conducted using Stata17 for Windows. The underlying mechanisms between the variables of the study were assessed using path analysis, a straightforward extension of multiple regressions used for modeling a set of hypothesized associations into a group of variables, including direct and indirect effects (mediational links) [65]. Path analysis is currently employed for both exploratory and confirmatory modeling, and therefore it allows theory testing and theory development [66]. In this work, path analysis was implemented through structural equation modeling (SEM), all parameters were free-estimated, and the maximum-likelihood method of parameter estimation was used. Because of the existence of multiple dimensions for the personality profile, a latent variable was defined by the observed indicators measured with the TCI-R scores, which allowed the data structure to be simplified and facilitated a more parsimonious fitting [67]. Additionally, with the aim to obtain a final parsimonious model and increase statistical power, an initial model that included all the potential associations between the variables was defined. Next, parameters without significant tests were deleted, and the model was respecified and readjusted. Adequate goodness-of-fit was evaluated for nonsignificant results in the chi-square test (χ2), root mean square error of approximation RMSEA < 0.08, Bentler’s Comparative Fit Index CFI > 0.90, Tucker–Lewis Index TLI > 0.90, and standardized root mean square residual SRMR < 0.10 [68]. 

## 3. Results

### 3.1. Description of the Participants and Distribution of the Genetic Measures 

Table 1 summarizes the distribution of socio-demographic and clinical variables in the study. Most participants were men (*n* = 134, 91.8%), married (*n* = 83, 56.8%) or single (*n* = 46, 31.5%), with low education levels (primary, *n* = 104, 71.2%), and employed (*n* = 100, 68.5%). Mean age was 40.2 years (SD = 12.5), mean age of GD onset was 34.2 years (SD = 11.9), and mean duration of the gambling problems was 13.7 years (SD = 8.6). The gambling preference with the highest prevalence was non-strategic (*n* = 125, 85.6%), followed by mixed gambling (non-strategic and strategic, *n* = 16, 11.0%). Slot machines were the games with the highest prevalence (*n* = 131, 89.7%), followed by bingo (*n* = 32, 21.9%), casino (*n* = 15, 10.3%), lotteries (*n* = 13, 8.9%), and cards (*n* = 10, 6.8%).

Table 2 summarizes the distribution of the genetic variables in the study, all of which were identified in the study by Solé-Morata et al. [36]. We included four single SNPs significantly related to GD according to different genetic models: (a) rs796189, the presence of genotype “AG/GG” (dominant model) and “AG” (overdominant model); (b) rs3763614, the presence of genotype “CC” (codominant and dominant models) and genotype “CC/TT” (overdominant model); (c) rs11140783, the presence of genotype “CC” (codominant model); and, (d) rs3739570 the presence of genotype “CC” (dominant model) and “CC/TT” (overdominant model).

We also analyzed three haplotypes significantly related to GD. Haplotype block 1 included SNPs rs6489630 and rs7956189 in the NTF3 gene, and haplotype “TG” was significantly related to an increase in the risk of GD (*p* = 0.045); haplotype block 2 included the SNPs rs4412435, rs10868241, and rs4361832 in the NTRK2 gene, and haplotype CAG (*p* = 0.048) was related to a decreased risk of GD. Finally, haplotype block 3, defined by the SNPs rs11140783 and rs3739570 among the NTRK2 gene, showed a significant association of haplotype CC with an increased risk of GD (*p* = 0.012) [36].

### 3.2. Path Analysis

Table 3 shows the association between the genetic variables. Because of the strong association between haplotype 1 with rs7956189 (Cramer-V = 1.00) and haplotype 3 with rs3739570 (Cramer-V = 0.737), both SNPs were excluded from the path analysis.

Figure 1 includes the path diagram with the standardized coefficients for the final model. Adequate goodness-of-fit was achieved: χ2 = 95.99 (*p* = 0.239), RMSEA = 0.027 (95% confidence interval: 0.001 to 0.054), CFI = 0.970, TLI = 0.963 and SRMR = 0.059. The global predictive capacity valued with the coefficient of determination was CD = 0.490.

The risk group for haplotype block 1 predicted higher debts due to gambling behavior, the group carrying the non-protective allele for haplotype block 2 predicted higher bets in the gambling episodes, and the risk group for SNP rs3763614 predicted worse psychopathological state. Within the SEM, no direct effect was observed for haplotype block 3 (this predictor only significantly correlated with haplotype 2), and SNP rs11140783 was excluded since there were no significant associations within the structural paths. No mediational links appeared between the genetic variables with socio-demographics (sex and age), personality and gambling-related measures.

The latent variable with the personality profile retained five TCI-R scales with significant coefficients (persistence and self-transcendence were excluded because of estimates with *p* > 0.05). Based on the coefficient’s values, higher scores in the latent variable are characterized by higher scores in novelty seeking and harm avoidance, and lower scores in reward-dependence, self-directedness, and cooperativeness. In addition, higher scores in this latent measure were associated with a worse psychopathological state, and it was also a mediational link in the relationships between sex and psychopathology.

The remaining significant associations in the diagram indicated that younger age was a predictor of higher bets per gambling episode and higher GD severity levels, while older age predicted higher debts. Finally, strategic gambling was associated with higher bets. 

Figure 2 includes the results of the path diagram of an additional SEM, which allows assessing specific associations between the genetic variables with each personality dimension (the TCI-R variables retained in the model because of significant structural coefficients were novelty seeking, harm avoidance, and cooperativeness). Adequate goodness of fit was achieved: χ2 = 76.16 (*p* = 0.123), RMSEA = 0.038 (95% confidence interval: 0.001 to 0.066), CFI = 0.907, TLI = 0.901, and SRMR = 0.062. The global predictive capacity was CD = 0.372. This new model showed a mediational link between genetic variables with personality and psychopathology: carrying the non-protective variant of haplotype 2 was related to a lower level in the cooperativeness factor, and lower values in this personality dimension were a predictor of greater psychopathological problems.

## 4. Discussion

Most of the evidence accumulated suggests that the development and maintenance of GD is associated with multiple biopsychosocial variables (e.g., genetics, psychological features, socio-demographics, etc.) [18]. Apart from being identified as vulnerability factors, elucidating the relationship between these genotypic and phenotypic features could allow to define vulnerability profiles of GD [28,54]. Despite its potential clinical and therapeutic implications, analyses based on the interaction between genotypic and phenotypic factors in GD are lacking since there are difficulties in defining the associations between variables under an integrative theoretical model as well as considering the complexity of the interplay by the measurement of direct and mediational links between them. 

Therefore, we aimed to assess whether NTF genetic variants previously associated with the development of GD [36] were also associated with GD severity in terms of their interaction with other different biopsychosocial variables. As we hypothesized, through SEM analysis, we found that the presence of these genetic variants predicted a higher GD severity, highlighting the mediational role of socio-demographic (e.g., sex, age), psychological (e.g., personality, and general psychopathology), and clinical variables related to gambling activity (e.g., age of GD onset). 

Overall, the socio-demographic and clinical characteristics of the patients agreed with those described in previous studies by our group [28,69]. The diagrams showed a strong correlation between the chronological age and the age of GD onset. Compared to individuals with a later age of onset, those who started gambling earlier in adolescence had more severe gambling problems (e.g., higher bets) [70]. In this line, younger age as well as an early age of GD onset have been bidirectionally linked to higher novelty seeking, which appears as a mediational factor in the pathogenesis and severity of GD [13,20]. Moreover, this profile (i.e., younger individuals with an earlier age of onset and higher novelty seeking) has been associated with lower levels of harm avoidance and cooperativeness, as we shown when we individually analyzed personality dimensions [20]. Notably, this trend is probably being accentuated by the advent of new technologies and changes in gambling availability [71]. On the other hand, a worse psychopathological state was related to older individuals with GD, which also predicted higher GD severity. Previous studies have reported higher general psychopathology and higher frequency of psychiatric comorbidities among older individuals with GD, previous to the development of this disorder [20,21]. Likewise, greater physical and psychological symptoms have also been described because of GD, possibly contributing to amplifying this maladaptive behavior [72]. According to our results, the multifactorial networks that mediate GD severity could be associated with the patients’ age. That is, higher GD severity was positively related to higher novelty seeking scores in younger patients, who were also characterized by early age of GD onset and higher bets. In contrast, a worse psychopathological state predicted GD severity among older individuals with GD, who tended to develop GD later in life but to take on more debt [21]. All these aspects support the heterogeneity of the disorder and the existence of subtypes based on different phenotypes and endophenotypes of GD patients [54,73,74]. 

Analyzing the contribution of sex on the underlying mechanisms of GD severity, a recent study showed that the complex links mediating GD severity were strongly related to sex [75]. Thus, while higher GD severity was directly related to a higher cognitive bias and a younger age of GD onset in men, GD severity was directly increased by younger age of onset, higher cognitive bias, and lower self-directedness among women. In this subgroup, lower socio-economic positions and higher levels in harm avoidance had an indirect effect on GD severity, mediated by the distortions related to the gambling activity. Going one step further, women generally seek treatment when they are older, and they commonly show higher levels of associated psychopathology, especially depression and anxiety, in comparison with men [76,77]. Along this line, although we failed to find a significant association between age and sex, our SEM showed that being female was positively associated with higher GD severity, and that this relationship appears to be mediated by a latent personality variant. Higher scores in this latent variant were translated into higher harm avoidance and lower self-directedness, among other personality measured dimensions. Furthermore, this latent variant predicted a worse psychopathological state. Considering both SEM analyses, we hypothesize that being a woman with higher levels of harm avoidance and a worse psychopathological state, despite lower scores in novelty seeking, would predict higher GD severity. Although our findings could partially be supported by the results of other studies, gender-specific studies focused on GD are scarce, and women with GD have been understudied. Therefore, further work is needed to better understand the differences between both sexes regarding GD severity and its biological correlates.

Regarding the psychological profile of patients with GD, the final SEM exhibited an already cited latent personality variable that directly predicted higher GD severity. Although our latent variable did not allow us to individually estimate the association of each personality dimension with the severity of GD, the whole picture was in agreement with previous research [21,78]. Regardless of the differences that may exist by sex or age, individuals with GD have been characterized as highly impulsive and usually show high scores in novelty seeking and harm avoidance [79,80,81]. In addition, a previous study by our group reported that novelty seeking and harm avoidance were positively and directly associated with GD severity, and also that sensitivity to reward and to punishment were meditational factors between these personality traits and GD severity [82]. In the case of general psychopathology, a worse psychopathological state has been reported among individuals with GD in comparison with the general population [83]. Interestingly, psychological distress has been considered a trigger for developing GD as a maladaptive strategy to regulate negative emotions [84]. Thus, the mediational role of these clinical features between biological variables and GD severity further emphasizes the complex and multifactorial etiology of GD [54,73,74]. 

While the results derived from the analysis of variables other than genetic traits are in line with the existing literature, one of the strengths of this work lies in its analysis of the associations between genetic risk variants for GD and the severity of the disorder. Regarding haplotype block 1, the risk group predicted higher debts due to gambling behavior in both SEM analysis. Although our study failed to find significant associations between debts that were caused by gambling activity and GD severity, other studies have reported a positive link [19]. 

Within the haplotype block 2, both SEM analyses suggested that the group carrying the non-protective allele “CAG” is indirectly and positively associated with higher GD severity. One the one hand, carrying the non-protective allele predicted higher bets in the gambling episodes, which was, in turn, correlated with higher severity of GD. In addition, younger age and strategic gambling were also related to higher bets per episode of gambling, emphasizing the existence of complex interrelations between the different variables. On the other hand, the non-protective allele was linked to lower cooperativeness, which predicted higher general psychopathology. These clinical interactions were in line with the results of previous studies. For example, a recent study based on cluster analysis showed that the group of patients with higher gambling severity was characterized by higher bets, younger age, early GD onset, a more dysfunctional personality profile, and greater psychopathological distress [19]. Although no direct links were observed for the risk group in haplotype block 3, it was significantly correlated with haplotype block 2, which are both found in the BDNF receptor’s gene (NTRK2). As they are known to be involved in some addictive-related disorders such as EDs [85,86], it is not surprising that genetic variants on this gene would be associated with GD severity. To summarize, the presence of haplotype block 2, interrelated with haplotype block 3, was associated with a more dysfunctional personality profile and a worse psychopathological state, which, in turn, had a direct link with GD severity. 

Finally, the presence of a particular CNTFR variant (rs3763614) was indirectly linked to GD severity through higher general psychopathology. At a molecular level, CNTF (i.e., CTNFR ligand) was shown to have an important effect on appetite and energy expenditure [87,88]. Thus, in line with previous studies, our findings suggest the existence of a common genetic pathway that could validate the NTF hypothesis role in some disorders related to impulsivity, such as EDs and GD [50]. 

Even though our study adds value to explorations of the interactions between already described phenotypic variables and genotypic features, genetic associations should be considered cautiously. Thus, the present work has an exploratory nature, and a causal relationship should not be established between these genetic markers and GD severity. Since the multifactorial pathogenesis of GD is complex, further functional analysis that includes larger samples, a wide range of genetic variants, and epigenetic influences, are needed to understand their biological impact on GD severity. However, the identification of genetic variants associated with the severity of a disorder, especially in the era of genomics, could be interesting from a clinical perspective to improve treatment approaches based on personalized medicine. 

A deeper knowledge about the complex pathophysiology of GD, as well as a better understanding of the biological factors underlying core clinical features in GD (e.g., genetics) and their modulatory interaction with other biopsychosocial variables could facilitate the identification of GD profiles with distinctive clinical implications in terms of, for example, the severity of the disorder. As a result, more individualized preventive and therapeutic strategies could be developed, which should be applied beginning in early clinical stages with the intent of ameliorating a more deleterious clinical course, overall, in patients with a more severe profile. Along this line, research based on biological therapeutic targets in GD is still preliminary and scarce, and no pyschopharmalogical drugs are officially approved for the treatment of GD [9,21]. Therefore, those studies that shed light on potential biological targets associated with the disorder would be opening the door to research into their treatment implications. According to previous literature, sharing risk factors between some disorders (e.g., addictive and impulsive related disorders) would not be uncommon [89,90]. This fact could also favor new avenues of treatment, such as through testing pyschopharmalogical drugs that are used in other disorders other than GD (e.g., substance use disorders), as these biological therapeutic targets may underlie transdiagnostic phenotypic features (e.g., impulsivity) [52,53].

## 5. Conclusions

The present study described an interesting vulnerability model based on potential interactions between genotypic (i.e., NTFs genes) and phenotypic features (i.e., socio-demographic, psychosocial, and clinical factors) through SEM analysis. These results provide a deeper insight into the biopsychosocial mechanisms underlying GD severity. According to the idea that GD is a complex multifactorial disorder, the presence of non-protective NTF genetic variants predicted a higher GD severity, with a mediational role of variables such as age, sex, personality traits, general psychopathology, and clinical features related to gambling behavior. Bearing the phenotypic heterogeneity of the disorder in mind, this work also sheds new light regarding the existence of vulnerability profiles whose identification would have potential applications in terms of diagnostic, preventive, and therapeutic approaches. However, future research based on longitudinal designs, with larger sample sizes and further functional analysis that includes not only the assessment of other NTF genetic variants but also other genetic targets, or the assessments of a wide range of biopsychosocial variables, are needed to consolidate these preliminary and exploratory results.

## Figures and Tables

**Figure 1 nutrients-15-00418-f001:**
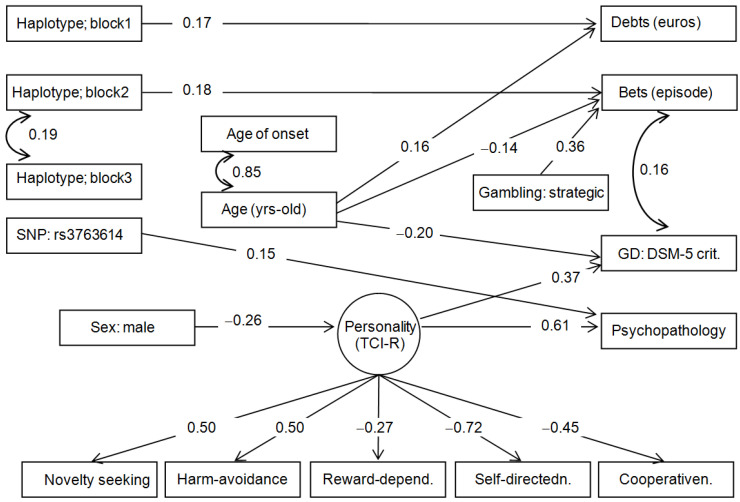
Path diagram: standardized coefficients (*n* = 146). Note: Only significant parameter estimates were retained in the final model. SNP: single nucleotide polymorphism. GD: Gambling Disorder.

**Figure 2 nutrients-15-00418-f002:**
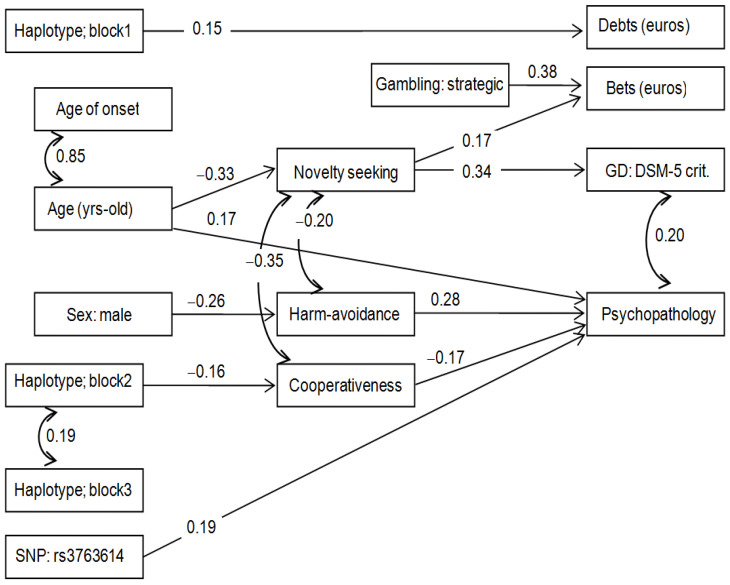
Path diagram: standardized coefficients (*n* = 146). Note: Only significant parameter estimates were retained in the final model. SNP: single nucleotide polymorphism. GD: Gambling Disorder.

**Table 1 nutrients-15-00418-t001:** Descriptive for the sample (*n* = 146).

Socio-Demographics		*n*	%
Sex	Women	12	8.2%
Men		134	91.8%
Civil status	Single	46	31.5%
Married—couple		83	56.8%
Divorced—separated		17	11.6%
Education level	Primary	104	71.2%
Secondary		36	24.7%
University		6	4.1%
Employment status	Unemployed	46	31.5%
Employed		100	68.5%
Age-onset-duration		Mean	SD
Chronological age (years-old)		40.2	12.52
Onset of gambling problems		34.24	11.89
Duration of gambling problems		13.72	8.60
Gambling activity (reason treatment)		*n*	%
Slot-machines		131	89.7%
Bingo		32	21.9%
Lotteries		13	8.9%
Casino		15	10.3%
Cards		10	6.8%
Preference (reason treatment)		*n*	%
Only non-strategic		125	85.6%
Only strategic		5	3.4%
Both (non-strategic and strategic)		16	11.0%
Bets		Median	IQR
Maximum euros per episode		400	300
Debts due to gambling activity		*n*	%
Yes		100	71.9%
No		46	28.1%

Note. SD: standard deviation. IQR: interquartile range.* Defined as the risk condition for the sample.

**Table 2 nutrients-15-00418-t002:** Distribution of the genetic variables in the study (*n* = 146).

		*n*	%		*n*	%
Block 1 *(SNPs: rs6489630, rs7956189)*				SNP: rs7956189 AA		
Haplotype	A	85	58.2%	AA	98	67.1%
	TA	13	8.9%	AG *	45	30.8%
	TG *	48	32.9%	GG *	3	2.1%
Block 2 *(SNPs: rs4412435, rs10868241, rs4361832)*				SNP: rs3739570 CC *		
Haplotype	CAA	2	1.4%	CC *	130	89.0%
	CAG	4	2.7%	CT	15	10.3%
	CGG	9	6.2%	TT	1	0.7%
	TGG *	131	89.7%	SNP: rs3763614 CC *		
Block 3 *(SNPs: rs11140783, rs3739570)*				CC *	137	93.8%
Haplotype	CC *	119	81.5%	CT	9	6.2%
	CT	14	9.6%	SNP: rs11140783 CC *		
	TC	13	8.9%	CC *	133	91.1%
				CT	13	8.9%

Note. SNP: single nucleotide polymorphism. * Defined as the risk condition for the sample.

**Table 3 nutrients-15-00418-t003:** Association of the genetic variables in the study (*n* = 146).

Cramer-V Values		2	3	4	5	6	7
1	Haplotype-1	−0.003	−0.042	1.000	−0.035	−0.124	−0.088
2	Haplotype-2	---	0.187	−0.003	0.315	−0.087	−0.106
3	Haplotype-3		---	−0.042	0.737	−0.122	0.656
4	rs7956189			---	−0.035	−0.124	−0.088
5	rs3739570				---	−0.090	0.044
6	rs3763614					---	−0.080
7	rs11140783						---

Note. SNP: single nucleotide polymorphism. Haplotype 1: SNPs: rs6489630, rs7956189. Haplotype 2: SNPs: rs4412435, rs10868241, rs4361832. Haplotype 3: SNPs: rs11140783, rs3739570.

## Data Availability

Individuals may inquire with the corresponding authors regarding availability of the data as there are ongoing studies using the data and to preserve patient confidentiality. Authors will consider requests on a case-by-case basis.

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
