# Peer review of "Underlying Mechanisms Involved in Gambling Disorder Severity: A Pathway Analysis Considering Genetic, Psychosocial, and Clinical Variables"

_nutrients, 2023, doi:10.3390/nu15020418_

Round 1
Reviewer 1 Report
Thank you for the opportunity to review this manuscript. The authors examined the underlying mechanisms implicated in Gambling Disorder severity. The topic is interesting, and I believe it could contribute to the literature on GD.
However, the paper is in a draft format and needs considerable reworking before it can be fully considered and suitable for publication.
My primary concern is the rationale. Overall, the theoretical rationale in the introduction needs to be considerably strengthened and more clearly structured. The authors must do a better job of explaining the importance and novelty of their study. What do the authors hypothesize and why? One thing that would help would be to re-write the introduction to be more integrative. Right now, it reads as a list of studies that support pieces of the model, and the motivation and justification for the full model are obscured.
Furthermore, the study's main result is analyzed in relation to the literature, the conclusion reflects the study's results, and the study's limits are comprehensive. However, given that no rationale was given, the discussion section needs more support. The authors suggest that the results provide valuable insights into the biological and clinical mechanisms underlying GD severity and shed new light on the endophenotypic profiles of individuals with GD. Can this empirical evidence be included in a theoretical construct, which may give these figures a meaning? The authors should attempt a discussion of the data that they have reported. Moreover, I would appreciate a discussion about clearer and specific clinical implications. It may be important to explore this implication and incorporate the findings of this study into the therapeutic procedure.
Once again, I appreciate your efforts in doing this study and writing up this article, and I wish you the very best.
Author Response
Point to point
Dear reviewer, we would like to thank your detailed review which highlights important points. An itemized point-by-point response is presented below. Modifications carried out to improve the paper were highlighted with yellow in the revised manuscript.
Reviewer 1.
Comments and Suggestions for Authors
Thank you for the opportunity to review this manuscript. The authors examined the underlying mechanisms implicated in Gambling Disorder severity. The topic is interesting, and I believe it could contribute to the literature on GD.
However, the paper is in a draft format and needs considerable reworking before it can be fully considered and suitable for publication.
My primary concern is the rationale. Overall, the theoretical rationale in the introduction needs to be considerably strengthened and more clearly structured. The authors must do a better job of explaining the importance and novelty of their study. What do the authors hypothesize and why? One thing that would help would be to re-write the introduction to be more integrative. Right now, it reads as a list of studies that support pieces of the model, and the motivation and justification for the full model are obscured.
Response: We are grateful for your suggestion. Now, we have reviewed and modify the Introduction section according to the reviewer’s suggestions to a better understanding of the rationale. In this line, we have also completed information about the justification, objectives, and hypothesis of this study.
Furthermore, the study's main result is analysed in relation to the literature, the conclusion reflects the study's results, and the study's limits are comprehensive.
Response: We thank your evaluation.
However, given that no rationale was given, the discussion section needs more support.
The authors suggest that the results provide valuable insights into the biological and clinical mechanisms underlying GD severity and shed new light on the endophenotypic profiles of individuals with GD. Can this empirical evidence be included in a theoretical construct, which may give these figures a meaning? The authors should attempt a discussion of the data that they have reported. Moreover, I would appreciate a discussion about clearer and specific clinical implications. It may be important to explore this implication and incorporate the findings of this study into the therapeutic procedure.
Response: Thank you for the appreciation. With the modification of the Introduction section, a more completed explanation of the justification, objectives, and hypothesis related to this study is now available. Therefore, we hope that the Discussion section will be better contextualized in that sense. Nevertheless, according to the reviewer’s comments, we have rewritten some parts of the Discussion section to construct a more clarified and integrative theoretical explanation for this empirical evidence. Besides, we included a larger discussion about the potential clinical and therapeutic implications of our results.
Once again, I appreciate your efforts in doing this study and writing up this article, and I wish you the very best.

Reviewer 2 Report
Subject: Manuscript etitled Underlying Mechanisms Involved in Gambling Disorder Severity: A Pathway Analysis Considering Genetic, Psychological, and Clinical Variables
The aim of this study was to assess the interrelations between a set of genetic, psychological, and clinical variables associated with Gambling Disorder (GD). This study described an interesting vulnerability model based on potential interactions between biological (i.e., neurotrophic factors (NTF) genes) and clinical variables, obtained through structural equation models analysis. According to authors it seems that NTF genetic variants previously associated with the development of GD were also associated with its severity. Thus, the presence of haplotype block 2, interrelated with haplotype block 3, was associated with a more dysfunctional personality profile and a worse psychopathological state, which, in turn, had a direct link with GD severity. On the other hand, having rs3763614 predicted higher general psychopathology and therefore, higher GD severity.
The sample was composed of 146 adult outpatients with Gambling Disorder in outpatient treatment. Several instruments were used (all validated), additional socio-demographic and clinical variables related to gambling were measured using a semi-structured, face-to-face clinical interview (previous work). Genetic data analyzed in the present study came from the analysis of previous work of the group.
Advantages
· Manuscript represents a contribution to the yet poorly researched field.
· According to authors this is the first clinical study examining the complex interactions between genetic, psychological, and socio-demographic factors in GD.
· The results provide valuable insights into the biological and clinical mechanisms underlying GD severity.
· The results might have a potential application in terms of diagnostic, preventive, and therapeutic approaches.
Weackness
· Preliminary results and further research is needed.
· Small sample (146 adults), 92% were males (only 12 females), sample of outpatient patients with GD in treatment: results cannot be applied to the whole population of patients with GD, female patients, other subgroups of patients (e.g. younger)
· Exploratory nature, a causal relationship should not be established between assessed genetic markers and GD severity (heterogeneity of the disorder).
This paper is very well organized, clearly written and professionally done, references are cited according to the rules of journal, the topic is important and accurate.
However please see lines: 111, 148, 389, 390 and complete
In conclusion, I recommend the Editor to publish this paper.
Author Response
Point to Point
Dear reviewer, we would like to thank your detailed review which highlights important points. An itemized point-by-point response is presented below. Modifications carried out to improve the paper were highlighted with yellow in the revised manuscript.
Reviewer 2.
Comments and Suggestions for Authors:
Subject: Manuscript entitled Underlying Mechanisms Involved in Gambling Disorder Severity: A Pathway Analysis Considering Genetic, Psychological, and Clinical Variables
The aim of this study was to assess the interrelations between a set of genetic, psychological, and clinical variables associated with Gambling Disorder (GD). This study described an interesting vulnerability model based on potential interactions between biological (i.e., neurotrophic factors (NTF) genes) and clinical variables, obtained through structural equation models analysis. According to authors it seems that NTF genetic variants previously associated with the development of GD were also associated with its severity. Thus, the presence of haplotype block 2, interrelated with haplotype block 3, was associated with a more dysfunctional personality profile and a worse psychopathological state, which, in turn, had a direct link with GD severity. On the other hand, having rs3763614 predicted higher general psychopathology and therefore, higher GD severity.
The sample was composed of 146 adult outpatients with Gambling Disorder in outpatient treatment. Several instruments were used (all validated), additional socio-demographic and clinical variables related to gambling were measured using a semi-structured, face-to-face clinical interview (previous work). Genetic data analysed in the present study came from the analysis of previous work of the group.
Advantages
- Manuscript represents a contribution to the yet poorly researched field.
- According to authors this is the first clinical study examining the complex interactions between genetic, psychological, and socio-demographic factors in GD.
- The results provide valuable insights into the biological and clinical mechanisms underlying GD severity.
- The results might have a potential application in terms of diagnostic, preventive, and therapeutic approaches.
Weakness
- Preliminary results and further research is needed.
- Small sample (146 adults), 92% were males (only 12 females), sample of outpatient patients with GD in treatment: results cannot be applied to the whole population of patients with GD, female patients, other subgroups of patients (e.g., younger)
- Exploratory nature, a causal relationship should not be established between assessed genetic markers and GD severity (heterogeneity of the disorder).
This paper is very well organized, clearly written and professionally done, references are cited according to the rules of journal, the topic is important and accurate.
However please see lines: 111, 148, 389, 390 and complete
In conclusion, I recommend the Editor to publish this paper.
Response: We thank you for reviewing this manuscript and the comments done. According to the reviewer’s suggestions we have checked and completed the referred lines.
- Line 111: To the best of our knowledge, this is the first clinical study examining the complex interactions between genetic, psychological, and socio-demographic factors in GD.
We have reviewed and modify the Introduction section according to the reviewers’ suggestions to a better understanding.
- Line 148: Additional socio-demographic and clinical variables related to gambling (e.g., type of problem gambling, age of onset, accumulated debts, average and maximum bets) were measured using a semi-structured, face-to-face clinical interview described elsewhere [41].
We have completed the described information in the text and added in Table 1 lacking data referred to debts and bets. Moreover, we specified that the collection of these data took place within the first session of evaluation. Now, the paragraph reads as follows:
Lines 204-210: “Additional socio-demographic (i.e., sex, chronological age, civil status, educational level, and employment status) and clinical variables related to gambling (i.e., age of GD onset and GD duration, type of gambling which motivated seeking-treatment and type of gambling modality regarding the preference for strategic gambling, non-strategic or mixed one, the presence of debts, and maximum bets) were measured within the first pre-treatment session of evaluation [64] (see also Table 1)”.
Furthermore, we have completed some parts of the Material and Methods section:
Lines 166-178:
“2.1. Participants and Procedure
The sample was composed of 146 adult outpatients with GD linked to the Behavioral Addictions Unit in the Department of Psychiatry at the University Hospital of XXX (XXX). All patients in this study fulfilled the DSM-5 criteria for GD [3], The recruitment took place between January 2005 and June 2006 [36]. They were evaluated the Behavioral Addictions Unit in the Department of Psychiatry at the University Hospital of XXX (XXX). The assessment consisted of two pre-treatment sessions. In the first session, a semi-structured clinical interview [55] was conducted by experienced psychologists and psychiatrists with a large clinical and research trajectory in the field of behavioral addiction such as GD. In the second session, psychometric assessments and biological sample to analysis genetic variables were obtained”. We had completed clinical assessments and biological samples analysis from all the participants included in this study.
- Lines 389-390: However, those are preliminary results, and further research is needed to deepen study these aspects.
We have reviewed and completed the Conclusions section according to the reviewer’s suggestions to a better understanding. Now it reads as follows:
Lines 476-491:
“5. Conclusions
The present study described an interesting vulnerability model based on potential interactions between genotypic (i.e., NTFs genes) and phenotypic features (i.e., socio-demographic, psychosocial, and clinical factors) through SEM analysis. These results provide a deeper insight into the biopsychosocial mechanisms underlying GD severity. According to the idea of being a complex multifactorial disorder, the presence of non-protective NTF genetic variants predicted a higher GD severity, with a mediational role of variables such as age, sex, personality traits, general psychopathology, and clinical features related to gambling behavior. Bearing the phenotypic heterogeneity of the disorder in mind, this work also sheds new light regarding the existence of vulnerability profiles, whose identification would have potential applications in terms of diagnostic, preventive, and therapeutic approaches. However, future research based on longitudinal designs, with larger sample sizes, further functional analysis including not only the assessment of other NTF genetic variants but also other genetic targets, or the assessments of a wide range of biopsychosocial variables are needed to consolidate these preliminary and exploratory results”.

Round 2
Reviewer 1 Report
Thank you for the opportunity to review this manuscript. I reviewed the revised paper in light of my previous comments. Overall, I do not have any additional comments, and I believe that most of the major issues have been appropriately addressed. The literature review and the discussion sections are considerably improved and the presentation of data and methods make the manuscript a lot more informative.
They have made substantial revisions and I believe that the revisions have significantly improved the quality of manuscript.